# Magma Plumbing System of Emeishan Large Igneous Province at the End-Permian: Insights from Clinopyroxene Compositional Zoning and Thermobarometry

**Jun-Hao Hu [1,*,†], Jing-Wen Liu [2,3,*,†], Tao Song [1] and Bai-Shun Shi [4]**

[1]   College of Urban and Rural Planning and Architectural Engineering, Guiyang University, Guiyang 550005, China; songtao198801@gmail.com

[2]   Center for Lunar and Planetary Sciences, Institute of Geochemistry, Chinese Academy of Sciences, Guiyang 550081, China

[3]   College of Earth and Planetary Sciences, University of Chinese Academy of Sciences, Beijing 100049, China

[4]   College of Earth Sciences, Jilin University, Changchun 130061, China; shibs17@mails.jlu.edu.cn

*   Correspondence: hujunhao2013@gyu.edu.cn (J.-H.H.); liujingwen@mail.gyig.ac.cn (J.-W.L.)

†   These authors contributed equally.

**Abstract:** The end-Permian Emeishan Large Igneous Province (ELIP) in SW China is widely accepted to have formed by mantle plume activities, forming voluminous flood basalts and rare picrites. Although many studies were performed on the petrogenesis and tectonic setting, the detailed conditions and processes within the magma chamber(s) remain unsolved. In this study, we studied the sector-/oscillatory-zoned clinopyroxene (Cpx) phenocrysts and performed Cpx-liquid thermobarometric calculation to constrain the physicochemical processes within the magma chambers. The results show that Cpx phenocrysts from the high-Mg basalts were crystallized at 4–27 (average 17) km, whilst those from the low-Mg basalt were crystallized at 0–23 (average 9) km depth. The sector and oscillatory Cpx zoning in the high-Mg basalts show that the magma had experienced undercooling and multistage recharge events in the deep-staging chamber(s). The magma replenishments may have eventually led to the eruption of high-Mg basalts, and magma ascent to the upper crust for further fractionation to form the low-Mg basalts and mafic intrusions.

**Keywords:** mineral zoning; clinopyroxene phenocryst; emeishan large igneous province (ELIP); magma plumbing system

## 1. Introduction

The research on the large igneous provinces (LIPs) is key to understanding continental uplift and breakup, and the associated environmental change, mass extinction, and magmatic Cu-Ni-PGE and Fe-Ti-V metallogeny [1]. For the end-Permian Emeishan LIP in SW China, several studies have shown that the flood basalts and picrites are genetically link to the regional magmatic Cu-Ni-PGE and Fe-Ti-V metallogeny [2–10]. Although the depth of the staging magma chamber(s) has been constrained through studying the volcanic rocks [11], the deep-chamber processes are yet to be well understood. Therefore, it is necessary to establish the ELIP magma plumbing system and its metallogenic link.

Compositional and textural zoning of minerals are not only influenced by the magmatic physicochemical changes [12], but also by the microscopic crystal-scale kinetics [13–15]. Therefore, the zoning characters of igneous minerals have been used to track magma chamber processes [16–22]. Different minerals have a different textural and compositional response to the surrounding magmatic

change, which is reflected in the formation or (partial) dissolution of the growth zones [14,23]. Clinopyroxene (Cpx) zoning has been widely documented in phenocrysts of mafic-intermediate volcanic rocks [24], and used to reveal the magma plumping system owning to the low lattice diffusion rate of the Cpx zoning [25–32]. Sector and oscillatory zoning of major and trace elements in pyroxenes, and their relations to magma evolution have been well established for synthetic and natural crystals [33–43]. Although the magma chamber conditions (e.g., pressure-temperature (P-T) and magma composition) also have strong effects on the formation of Cpx compositional zoning, sector zoning is closely related to the kinetic effects which may indicate the rate of magma ascent [14,34,37,44,45]. Meanwhile, Cpx oscillatory zoning can also reveal the physiochemical conditions and their changes/fluctuations during magma crystallization, including magma convection and replenishment, and crystal-melt interactions along the crystal boundary layer [18,38,39,46–50]. Moreover, the Cpx-liquid thermobarometer has relative advantages (e.g., the range of the application and precision in P-T calculation) in thermobarometric calculation compared to the olivine- or plagioclase-based ones [51]. Therefore, in this study, we investigate the compositional zoning of the Cpx phenocrysts from the ELIP high-Mg and low-Mg flood basalts. By using electron probe microanalysis (EPMA) and mapping of major elements, and Cpx-liquid thermobarometry, we established a reliable magma plumbing system for the ELIP staging chamber.

## 2. Geological Background

The ELIP exposed in southwestern China and northern Vietnam contains a wide variety of magmatic rocks including mafic-ultramafic intrusions, granitoids, flood basalts and picrites (Figure 1b) [52]. Geochemical studies suggested that the intrusive and extrusive rocks in the ELIP are genetically linked [53–55]. The volcanic succession overlies the Lower Permian Maokou Formation (Fm.) limestone, which in turn is overlain by the Upper Permian Xuanwei Fm. sandstone. Radiometric dating of mafic-ultramafic intrusions constrained the ELIP emplacement at ~260 Ma, coeval with the end-Guadalupian mass extinction [9,10,56–58].

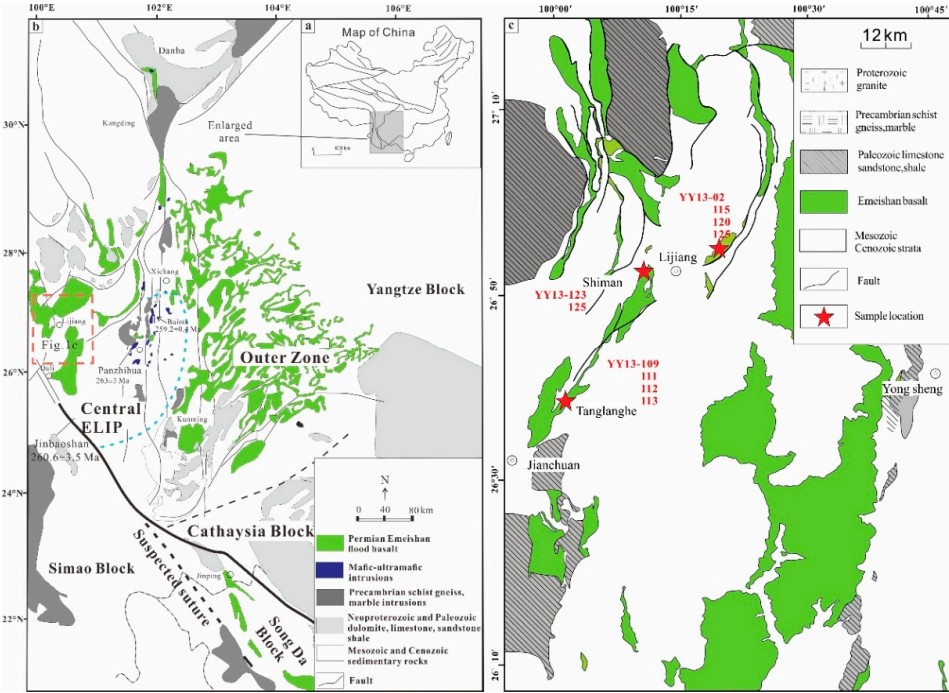

**Figure 1.** (**a**) Geologic map showing the simplified tectonic framework of China. (**b**) Distribution of the late-Permian Emeishan flood basalts and coeval mafic intrusions, modified after Yu et al. (2017) [6]. (**c**) Sample locations in this study.

The ELIP volcanic succession is exposed across an area of over $3 \times 10^5$ km$^2$ [52,57]. Based on the whole-rock Ti/Y ratios, the ELIP volcanic rocks have been classified into the high-Ti and low-Ti series [52]. The high-Ti basalts (Ti/Y > 500) are widespread across the ELIP, whereas the low-Ti basalts (Ti/Y < 500) are largely confined in the ELIP central zone [59]. Picrites are locally exposed in the western (Dali, Lijiang, Binchuan, Yongsheng, Ertan, Muli) and southern (Jinping-Song Da) parts of the ELIP, where they (3–50 m thick) are found interbedded with basaltic flows [6,55,60,61]. The high-Ti picrites have higher Nb contents than their low-Ti counterparts [55,62]. The two major ore deposit types in the ELIP, i.e., magmatic Fe-Ti-V oxide and Ni-Cu-PGE sulfide, were suggested to be genetically related to the high-Ti and low-Ti magma series, respectively [5,61].

## 3. Sampling and Analytical Techniques

Ten samples were collected from Tanglanghe and Shiman along the Dali-Lijiang region of the ELIP (Figure 1c). Systematic sampling across the volcanic stratigraphy was not possible due to the poor exposure and severe weathering/alteration of the lava flows, yet the samples collected still retain clear and fresh Cpx phenocrysts. Whole-rock major element compositions were measured by an X-ray fluorescence spectrometer (XRF) at the ALS Laboratory (Guangzhou) using fused lithiumtetraborate glass pellets. Analytical precision determined on the standard SARM-45 is generally 1–5%. The equilibrium melt composition is critical for accurate mineral-melt thermobarometric P-T calculation. Therefore, the volcanic groundmass was separated and analyzed to represent the melt equilibrated with the phenocrysts. For the analysis, 2–3 kg of the least-altered rock sample was crushed into 40–60 mesh in an agate mortar, and the Cpx phenocrysts were carefully picked out under a binocular microscope. The separated groundmass (~5 g) was milled into 200 mesh. Major elements of the groundmass were measured by XRF on fused glass disk at the Institute of Geology and Geophysics, Chinese Academy of Sciences. Analytical uncertainties are 1–3% for elements of over 1 wt% and ~10% for elements of below 1 wt%. The trace element contents of the whole-rock were measured by solution inductively coupled plasma mass spectrometry (ICP-MS) using the technique of Qi et al. (2000) [63] at the Institute of Geochemistry, Chinese Academy of Sciences (IGCAS). The standard samples (e.g., BCR-2, BHVO-2) were used for analytical quality control, with precision for most elements analyzed better than 5%. The major element composition data for the whole-rock and groundmass samples were back-calculated to 100 wt% anhydrous values (Supplementary Tables S1 and S2).

Major element contents of the Cpx phenocrysts were analyzed with a JEOL JXA-8100 electron microprobe at the State Key Laboratory for Ore Deposits Geochemistry, IGCAS. The measurement was performed with a 15-kV acceleration voltage, 20 nA beam current, and 3 μm spot diameter. The detection limit for major elements is 0.01 wt%, and the analytical reproducibility is within 2%. EMPA results are listed in Supplementary Table S3. Quantitative wavelength-dispersive spectrometric (WDS) analysis for the zoned Cpx phenocryst was conducted using an accelerating voltage of 15 kV, a beam current of 20 nA, and a spot size of 1 μm.

## 4. Results

### 4.1. Petrography

The samples YY13-109, 111, 112, 113, and 123 are more porphyritic with 20–50% olivine phenocrysts and <5% Cpx phenocrysts. The olivine phenocrysts are 2–3 mm large and commonly serpentinized along its margin and micro-fractures. Some olivine crystals also contain small euhedral chromite inclusions (Figure 2a). The Cpx phenocrysts are usually euhedral and around 2 mm in size, and are generally fresher than the olivine phenocrysts (Figure 2b–d). The fine-grained groundmass is composed of pyroxene, plagioclase and minor Fe-Ti oxides, among which some pyroxene and plagioclase are partially altered to chlorite, talc, and clay minerals. The samples of YY13-2, 115, 120, 122, 125 are less porphyritic with <5% Cpx phenocrysts. The Cpx phenocrysts are euhedral and generally ~2 mm in size (Figure 2e–h). Some crystals show compositional zoning under the microscope (Figure 2f).

Rare chromite inclusions were also identified in the Cpx phenocrysts (Figure 2h). The microcrystalline groundmass contains mainly pyroxene and plagioclase, and minor Fe-Ti oxides.

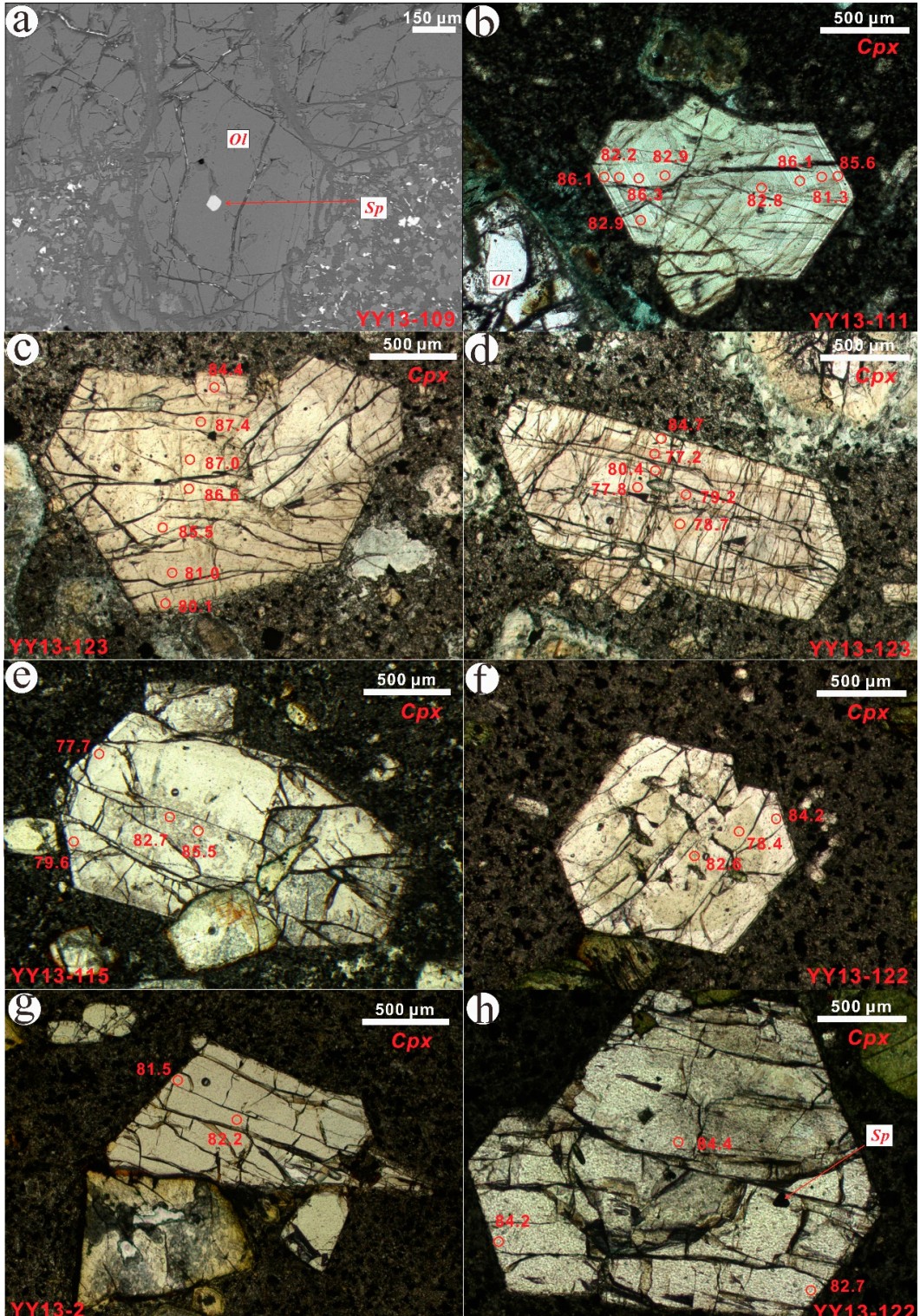

**Figure 2.** Representative BSE and microphotographs (plane-polarized light) of phenocrysts in studied samples. (**a**) Olivine phenocryst in high-Mg basalt. (**b–d**) Cpx phenocryst in high-Mg basalt. (**e–h**) Cpx phenocryst in low-Mg basalt. (The red square denotes the EPMA analytical spot for major elements. The number beside the red square represents the Mg# [MgO/(MgO + FeO)]. Ol = olivine, Cpx = clinopyroxene, Sp = spinel).

## 4.2. Whole-Rock Major and Trace Element Compositions

The major oxide contents of the analyzed samples are listed in Supplementary Table S1 and shown in Figure 3 and Supplementary Figure S1. We classified the samples into the high-Mg (MgO > 18 wt%) and low-Mg basalt (MgO < 18 wt%) groups using the standard suggested by Le Bas et al. (2000) [64]. Both groups are high-Ti basalt with Ti/Y > 500 (Supplementary Table S1). According to the TAS and Nb/Y vs. Zr/Ti diagrams (Figure 3), the samples are calc-alkaline to tholeiitic. The high-Mg basalts have higher MgO contents (18.12–24.82 wt%) and $Mg^\#$ $((Mg/Mg + Fe_{total}) \times 100 = 74 - 79)$ than the low-Mg basalts (MgO = 8.77–12.33 wt%; $Mg^\#$ = 56–65) (Supplementary Table S1, Supplementary Figure S1). The $TiO_2$ and $Al_2O_3$ contents correlate negatively with $Mg^\#$, while the Ni and Cr contents display positive correlations with $Mg^\#$, respectively (Supplementary Figure S1).

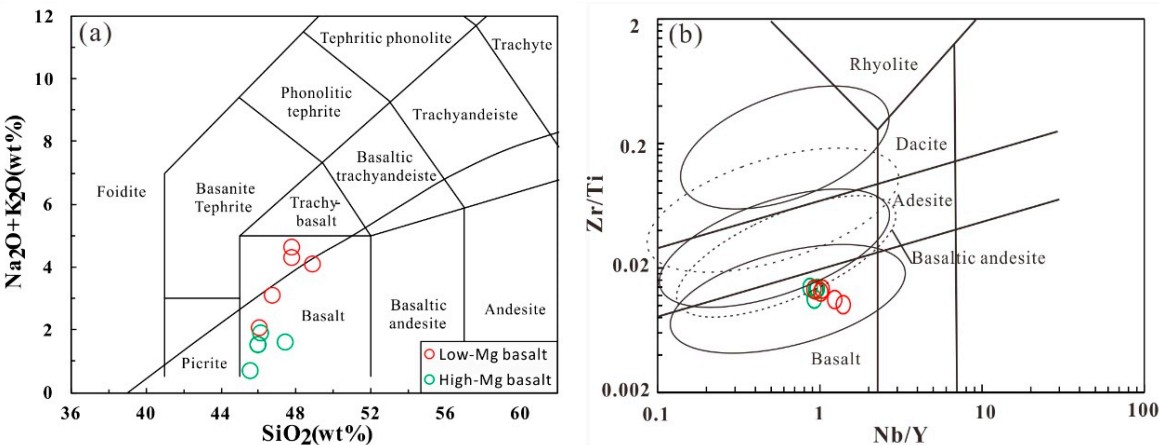

**Figure 3.** (**a**) Total alkali ($Na_2O + K_2O$) versus $SiO_2$ (TAS); (**b**) Nb/Y versus Zr/Ti(after Le Maitre et al. [65]. The diagram for sub-alkaline rocks is from Pearce 1996 [66]).

The high-/low-Mg basalt groups have distinctive incompatible element compositions. The high-Mg basalts are marked by lower REE concentrations than the low-Mg basalts (Supplementary Table S1). In the chondrite-normalized REE diagrams, all the basalts exhibit steeply right-sloping patterns with distinct LREE enrichment (Figure 4a). In the primitive mantle-normalized multi-element diagrams, both high-/low-Mg basalts show OIB-like patterns except for the LILE (e.g., Rb, Ba), and display a negative anomaly in Sr and positive anomaly in Ti (Figure 4b). The analyzed basalts have much higher Nb/U ratios (high-Mg basalt: 42.9–46.9; low-Mg basalt: 49.9–71.4) than that of the average bulk continental crust (12.09, [67]) (Supplementary Table S1). Moreover, the Th/Nb ratio of the basalt samples (0.07–0.10) is comparable to that of the average OIB (0.083, [68]) (Supplementary Table S1).

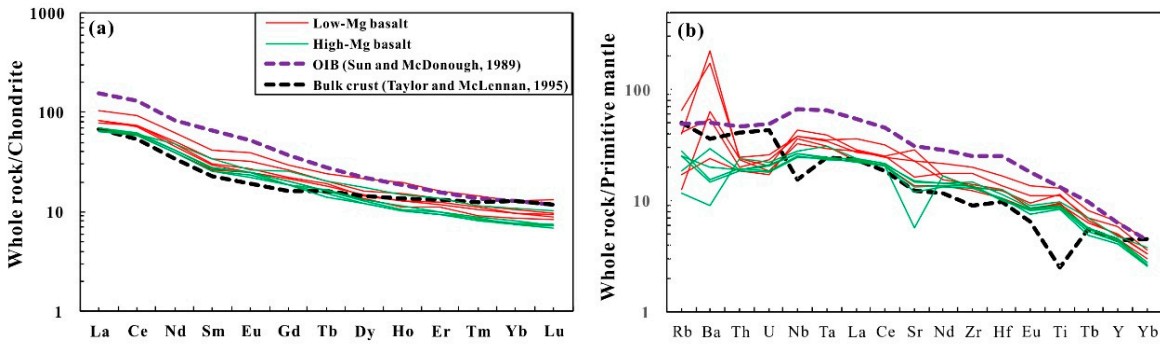

**Figure 4.** (**a**) Chondrite-normalized REE patterns of the low-Mg and high-Mg basalt. (**b**) Primitive mantle-normalized diagrams of incompatible elements for the low-Mg and high-Mg basalt. (The Chondrite and Primitive mantle values from Sun and McDonough (1989) [68]).

## 4.3. Clinopyroxene Phenocryst Compositions

The Cpx phenocrysts from all the basalt samples belong to the augite-diopside-solid solution series in the Wo-En-Fs diagram (Figure 5). The Cpx compositional variation is generally straightforward, as illustrated in the $Mg^\#$. The Cpx phenocrysts show positive $Mg^\#$ vs. $Cr_2O_3$ trend, and negative trends between $Mg^\#$ and $TiO_2$, $Al_2O_3$ (Supplementary Figure S2). Moreover, the Cpx phenocrysts in high-Mg basalts have lower $Al_2O_3$ contents (0.51–2.29 wt%) than those in low-Mg basalts ($Al_2O_3$: 1.71–7.51 wt%) (Supplementary Figures S2 and S3).

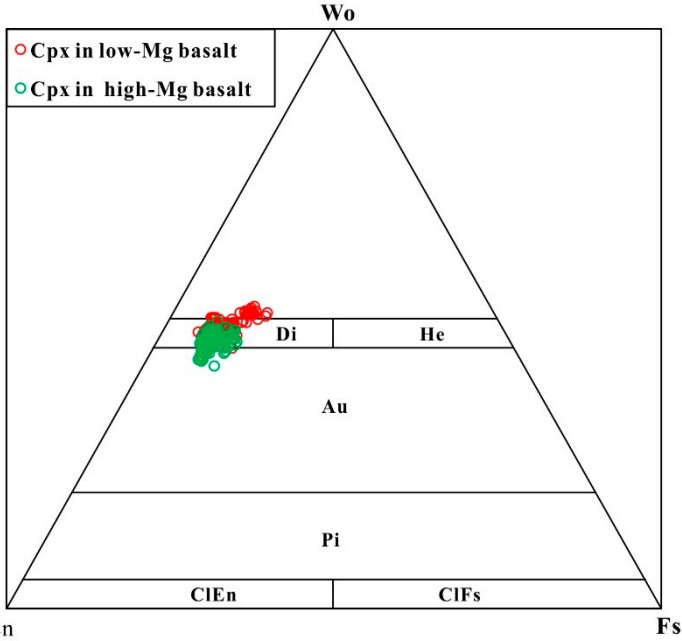

**Figure 5.** The pyroxene composition diagram (Di—diopside, Hehedenbergite, Au—Augite, Pi—Pigeonite, ClEn—clinoenstatite, ClFs—clinoferrosilite, Morimoto, 1988 [69]).

The Cpx phenocrysts are mainly normal-zoned, which is characterized by decreasing $Mg^\#$ and Cr from the core to rim, although rare reverse-zoned Cpx grains were also found (Supplementary Table S3). Clear oscillatory and sector zoning were observed in the Cpx phenocrysts from the high-Mg basalt samples (e.g., YY13-111 and 123). The Cpx phenocryst cut perpendicularly the c-axis and off the center, revealing clear zonation of Mg, Fe and Al contents (Figure 6b–d). The compositional profile (from A to B) of this Cpx phenocrysts shows positive correlations among the $Al_2O_3$, $TiO_2$ and FeO contents, which are negatively correlated with the $SiO_2$, MgO and $Cr_2O_3$ contents (Figures 6b–d and 7). Especially, the {1 0 0} has relative high contents of $Al_2O_3$, $TiO_2$ and FeO, and lower contents of $SiO_2$ and MgO than those in {-1 1 1} (Figures 6b–d and 7). In the sample YY13-123, the Cpx phenocrysts are cut perpendicularly to the b-axis and off the center. The hourglass sector ({-1 1 1}), which is located in the inner part of the crystal, has higher MgO and $SiO_2$ contents but lower $Al_2O_3$, $TiO_2$, and $Cr_2O_3$ contents than the prism sector ({0 1 0}) (Figures 6f–i and 8). The outer part of this Cpx phenocryst show oscillatory zoning, in which the $Cr_2O_3$ content correlates positively with MgO but negatively with $Al_2O_3$ and $TiO_2$ (Figures 6f–i and 8; Supplementary Figure S3). Olivine phenocrysts only occur in the high-Mg basalts, and their compositions are given in Supplementary Table S3.

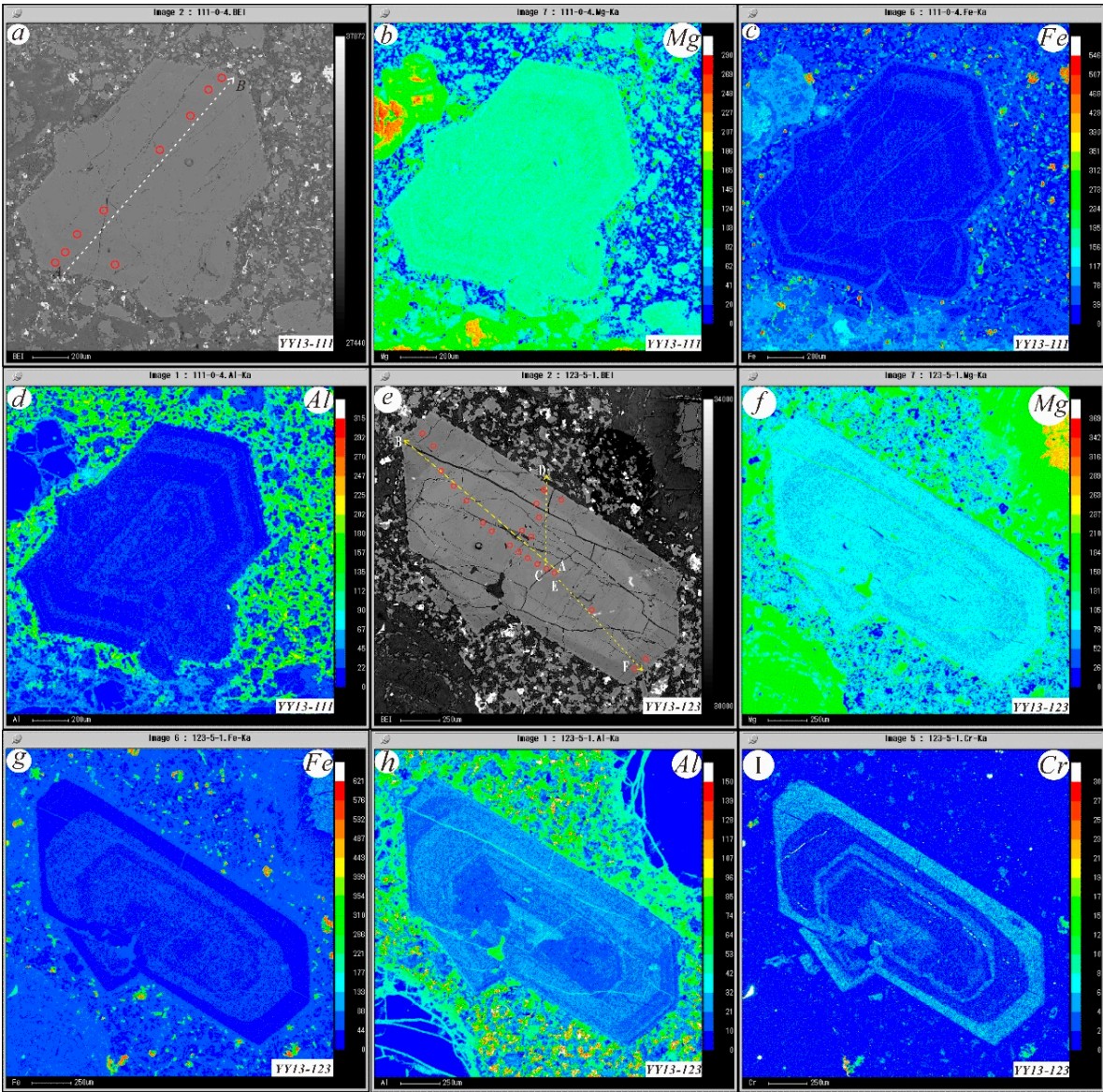

**Figure 6.** (**a**,**e**)The back-scattered electron images of the Cpx crystal from the high-Mg basalt sample YY13-111 and YY13-123, respectively, the detail data of the white dotted arrows from A to B are presented in Figure 7, Figure 8 and Supplements Figure S3. (**b**–**d**) and (**f**–**i**) qualitative WDS maps of Mg, Fe, Al and Cr.

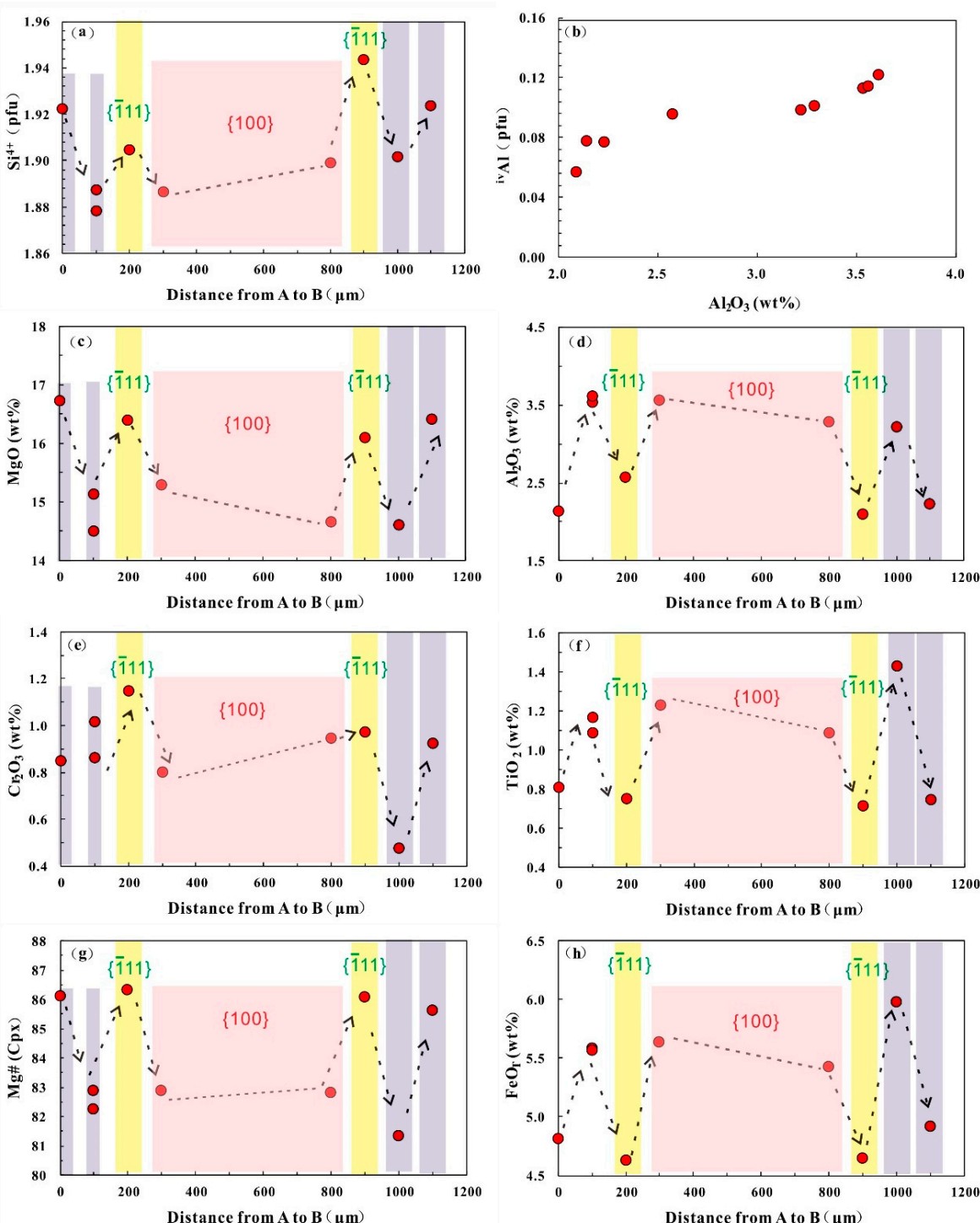

**Figure 7.** Elemental profiles (A to B) for the zoned Cpx phenocryst of the high-Mg basalt sample YY13-111. Profile analysis shows oscillatory zonings in the elements of Si, Mg, Cr, Ti and Al. The MgO and $Cr_2O_3$ show strong negative correction with the $Al_2O_3$, $TiO_2$ and FeO. The {100} shows $Al_2O_3$, $TiO_2$ FeO enrichment compared with the {-100}. (**a–h**) Major elements variations along the line (from A to B).

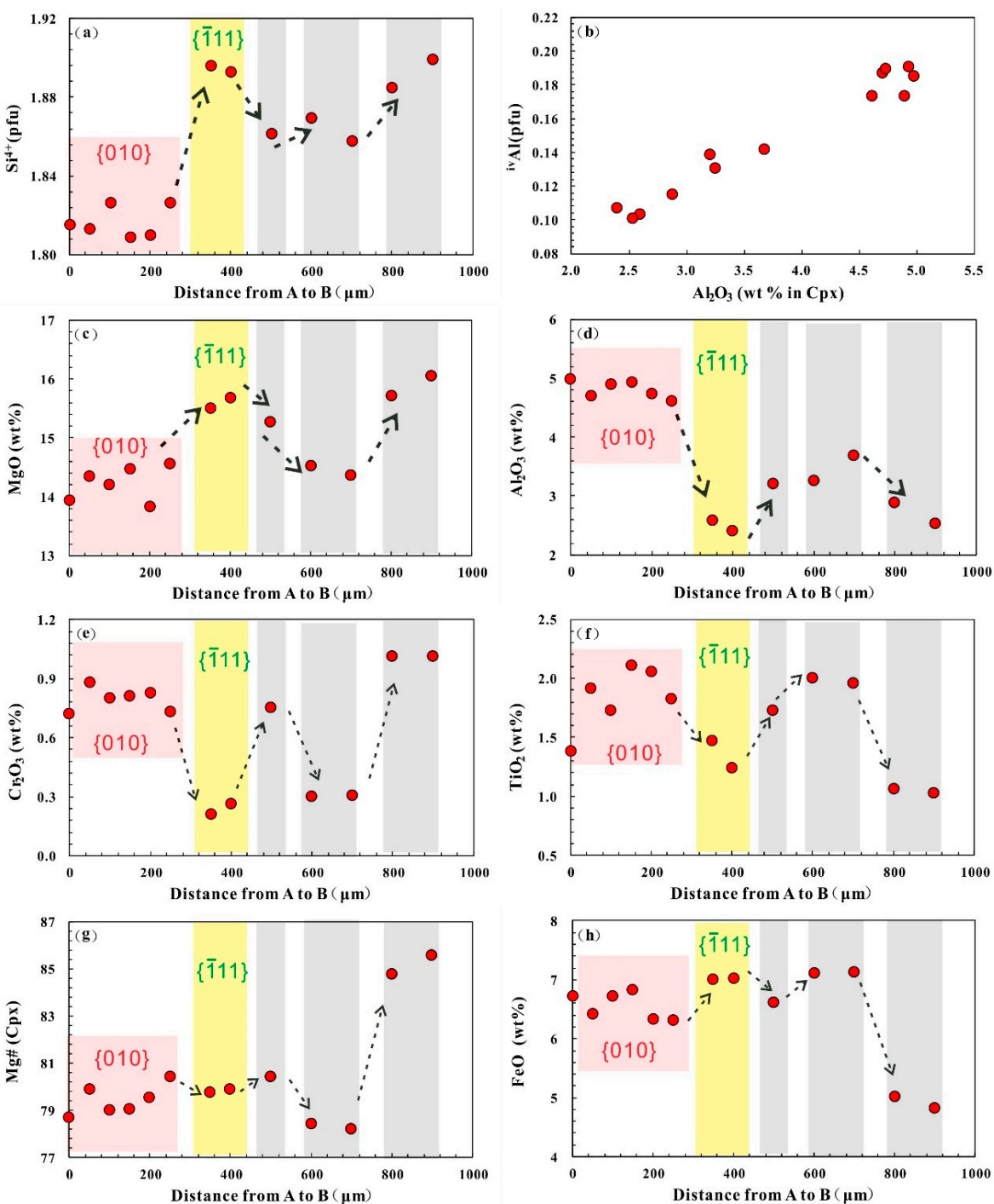

**Figure 8.** Elemental profiles (A to B) for the zoned Cpx phenocryst of the high-Mg basalt sample YY13-123. Profle analysis shows oscillatory zonings in the elements of Si, Mg, Cr, Ti and Al. The MgO and $Cr_2O_3$ show strong negative correction with the $Al_2O_3$, $TiO_2$ and FeO. The {010} shows $Al_2O_3$, $TiO_2$ FeO enrichment compared with the {-100}. (**a**–**h**) Major elements variations along the line (from A to B).

## 5. Discussion

### 5.1. Petrogenetic Implications from Zoned Clinopyroxene Phenocrysts

Different types of Cpx compositional zoning (e.g., sector, oscillatory, normal, reverse) have been studied in geological and synthetic samples [14,24,26,37]. Fractional crystallization in a closed system

and usually forms euhedral normal-zoned phenocrysts, and hence complex zoning (e.g., sector and oscillatory) mainly reflect changes in the magmatic environment [18,49,70].

In terms of concentric compositional zoning, variation occurs radially from the crystal core in response to the evolution of magmatic conditions and/or melt compositions [12,71]. Sector zoning, in contrast, requires faces of the same crystal to grow in a common temperature, pressure, and melt environment, yet incorporating elements in different concentrations and forming an anisotropy of element partitioning among prominent crystal faces [15]. Crystallographic studies indicate that crystals may grow with rates along different crystallographic orientations, which could result in element enrichments in one set of a sector and depletions in another [33,72,73]. Experimental studies highlight the kinetic effects induced by undercooling on the formation of sector zoning [34,44]. In these experiments, under a low degree of undercooling ($\triangle$T = 13–18 °C), the crystallization kinetics lead to the relative enrichment of incompatible cations (e.g., Al and Ti) in the {h k 0} sector compared to the {-1 1 1} sector. Under a higher degree of undercooling ($\triangle$T > 25–45 °C), Al and Ti are concentrated in the {-1 1 1} sector, and the crystals usually develop hopper to dendritic morphology. The above phenomena have been documented in natural Cpx crystals [14,15]. In this study, sector zoning is found in the inner part of the Cpx phenocrysts, which exhibits Al and Ti enrichments (Si and Mg depletions) in the sectors of {1 0 0}, {1 1 0}, and {0 1 0}, and depletions (Si and Mg enrichments) in the {-1 1 1} sector (Figures 6–8; Supplementary Figure S3). The reverse correlations between (Si, Mg) and (Al, Ti) in these sectors possibly reflect coupled substitution and/or charge balance compensation, which can be expressed as: $[Si^{4+} + Mg^{2+}]_{\{-1\,1\,1\}}$ — $[Al^{3+} + Ti^{4+}]_{\{h\,k\,0\}}$. These characters of sector zoning resemble those obtained from natural Cpx analyses and experimental studies [15,34]. The sector zoning infers that the crystal growth resulted from sluggish crystallization kinetics driven by mild undercooling during the early crystal growth stage.

The elements (e.g., Mg, Fe, Al, Ti, and Cr) in the outer part of the sector zone show oscillatory partition (Figures 6–8; Supplementary Figure S3). Although the interplay of crystal growth and chemical diffusion within the boundary layer could form fine and flat oscillatory zoning (<15 μm thick) [24,46–48,74], it is inconsistent with our observation that the oscillatory zones are ~50 μm thick (Figure 6). The oscillatory zoning could be related to the magma convection in the chamber, which may have resulted from periodic degassing processes in shallow reservoirs [49,75]. However, such degassing was unlikely at the crystallization depths inferred from previously published [11] and our new barometry data. The $Cr^{3+}$ shows oscillatory zoning through the whole crystals and is not portioned amongst sectors may be owning to the less ability to facilitate charge-balanced configurations as previously suggested [14] (Figures 7 and 8). The intriguing behavior of Cr in the system deserves consideration due to its perfect oscillatory variations (Figure 6i, Figure 7e, and Figure 8e). No experimental studies had shown that low-Cr pyroxene can be crystallized from high-Cr magma, as Cr is strongly partitioned into Cpx regardless of variation in intensive parameters (e.g., temperature and pressure) [76,77]. Therefore, the distinct oscillatory Cr content variation from the core to the rim in the Cpx crystals possibly indicate magma compositional changes led by repeated magma replenishment [12,14,18,49,77]. In this study, the oscillatory zones in the Cpx phenocrysts display positive correlation of Cr with $Mg^{\#}$, but negative ones with Al and Ti (Figures 7c–f and 8c–f; Supplementary Figure S3), The partition characters of these elements consist thermodynamic principles under near-equilibrium conditions. The geometrical perfection of crystal faces and distinct concentric zoning from core to rim (Figure 6), highlight the polyhedral crystallization under conditions approaching equilibrium [14,44]. Therefore, the sector and oscillatory zoning in the Cpx phenocrysts of high-Mg basalts indicate that the magma may have experienced a relatively low-degree undercooling during its early growth stage and eventually erupt due to the multistage magma replenishments.

*5.2. Clinopyroxene-Melt Thermobarometry*

Olivine and Cpx are the main phenocrysts in the high-Mg and low-Mg basalts, respectively. No reliable olivine–liquid geobarometry is currently available due to its strong temperature sensitivity

limit [51]. Moreover, the Cpx-melt thermobarometer is more accurate in determining the P-T conditions for phenocryst crystallization and has been widely used to study the magma plumbing system of volcanic rocks [22,78–81]. Therefore, our Cpx P-T calculation has adopted the model developed by Putirka (2008) [22]. Equation (30) for pressure and Equation (33) for temperature were used, with the compositions of coexisting liquid and Cpx compositions used for the basaltic compositions. Equation (30) was calibrated with a wide experimental dataset, and the estimated standard errors are ±1.6 kbar. Equation (33) is developed based on global calibrations using experiments conducted at P < 70 kbar, and the standard estimated is ± 45 °C. The geobarometer developed by Nimis and Taylor (2000) [82] has relatively low-temperature errors (±30 °C) and is also used here for result comparison. Due to the absence of the mineral melt inclusion for calculating the $H_2O$ contents in the ELIP, we assumed that the $H_2O$ is 0.7% for basalts, similar to that of non-subduction-related plumes [11].

By using the $K_D(Fe-Mg)^{cpx-liq} = 0.27 \pm 0.03$ [83] and Na-Ca-Al composition [45,84] for the basaltic system, respectively (Supplementary Figure S4). It is known that all the Cpx phenocrysts, including those with the highest $Mg^{\#}$ in the high-Mg basalts, are not in equilibrium with their groundmass. This could be thought as that the Cpx phenocrysts in the high-Mg basalts were crystallized after most olivine, and the groundmass still had some residual olivine phenocrysts. To test whether it is possible to acquire a liquid in equilibrium with the observed Cpx, we removed the rare olivine from the groundmass until $K_D(Fe-Mg)^{cpx-liq} = 0.27$, and then check whether the amount of olivine removed is equal or less than the measured olivine content. We found that about 10–15% of the olivine was removed. This, plus the number of olivine phenocrysts picked out under the binocular microscope, are less than the observed modal amount of olivine in the whole-rock samples. Supplementary Figure S4b shows that most of the Cpx phenocryst rims of the high-Mg basalts are in equilibrium with the groundmass, in which extra olivine phenocrysts were removed. Therefore, only the data of Cpx rim plotted in the equilibrium field (Supplementary Figure S4) are considered in the P-T estimation.

Our P-T calculation results suggest that the crystallization of the Cpx rims in the low-Mg basalts occurred under 0.44 to 6.02 (3.16 ± 1.53) kbar and 1109.5 to 1141.8 (1121.93 ± 9.86) °C. The Cpx phenocryst crystallization in the high-Mg basalts occurred under higher pressure (2.91 to 7.55 (5.71 ± 1.46) kbar) and temperature (1181.79 to 1236.37 (1210.67 ± 16) °C) (Figure 9a; Supplementary Table S4). As the geobarometer suggested by Nimis and Talyor (2000) [82] is more applicable to the high-Mg rocks, the results suggest narrow temperature variation in the high-Mg basalts (1157.88 to 1210.34 (1196.43 ± 11) °C) (Figure 9b). Based on geophysical data, Jiang et al. (2012) [85] suggested that the upper, middle, and lower crustal boundaries in the western Yangtze craton are located at 16, 33, and 55 km deep, respectively. Taking the standard error of thermobarometry into consideration, the estimated Cpx crystallization depths for the high-Mg basalts are 4–27 (average 17) km, corresponding to the upper to middle Yangtze crust, whilst those for the low-Mg basalts are 0–23 (average in 9) km, corresponding to the upper Yangtze crust.

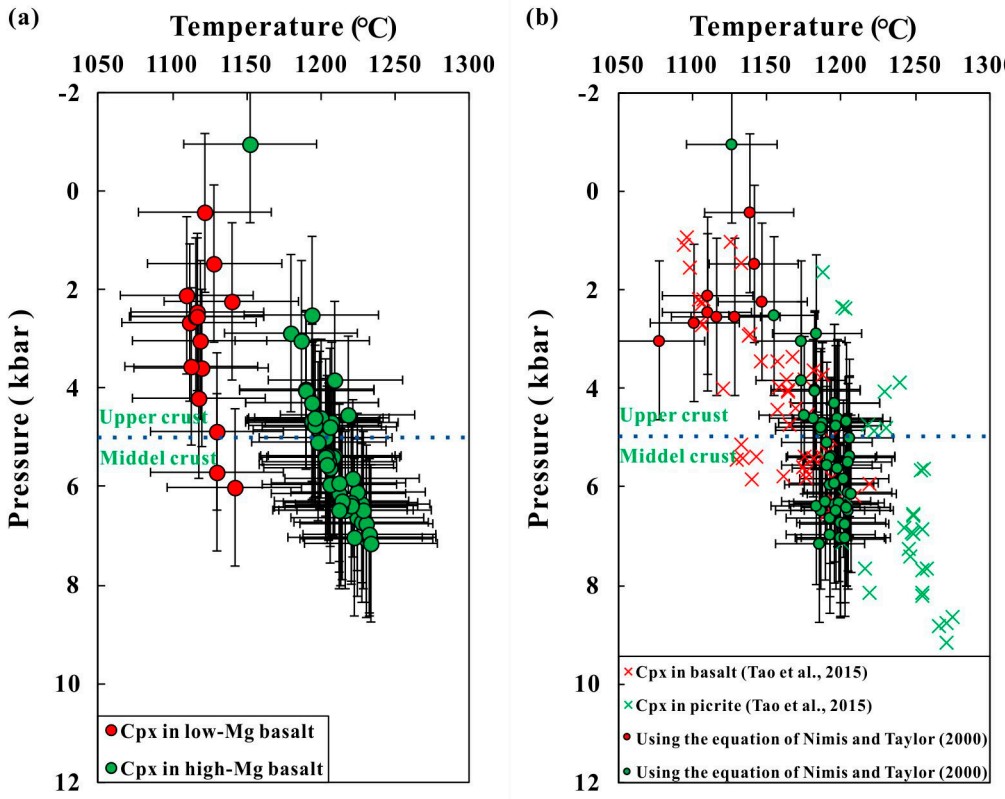

**Figure 9.** Cpx equilibrium pressures and temperatures calculated by chemical thermobarometry. Each sign denotes data point from Cpx phenocryst-matrix pair. (The error bars in a and b show the SEE of the geothermometer methods from Putirka, 2008 [22] and Nimis and Talyor (2000) [82], respectively). (**a**) P-T calculation based on the model of Putirka, 2008 [22]; (**b**) P-T calculation based on the model of Nimis and Talyor (2000) [82].

*5.3. Magma Plumbing System in the ELIP*

It is generally agreed that the ELIP was emplaced in a short duration [6,86]. Together with the published pressure data from the high-Ti volcanic rocks [11] (Figure 9b), we support that a series of magma chambers were present across the middle to upper crust. The fractures likely provided the conduits for the magma ascent and ponding for the fractional crystallization [78,87,88]. Previous studies also suggested that the regional fractures and density contrast (between the magma and wallrocks), have significant effects on forming the ELIP magma plumbing system. Meanwhile, magma replenishments have likely played key roles in generating the ELIP magma plumbing system and eventually led to volcanic eruptions [49]. The magma recharge events in the ELIP are supported by the presence of magmatic cycles and massive Fe-Ti oxide layers in the mafic-ultramafic intrusions (e.g., Panzhihua intrusion) [89].

As above-mentioned, the progressive $Al_2O_3$ increase with decreasing MgO indicates the lack of significant plagioclase fractionation from the high-Mg to low-Mg basalts (Supplementary Figure S1), the decrease in compatible elements (such as Cr and Ni) with fractionation indicates extensive pyroxene and olivine crystallization (Supplementary Figure S1). Varying degrees of crustal assimilation are common for mantle-derived magmas ascending through continental crust [90,91]. However, trace element compositions of the high-Mg and low-Mg basalts suggest that their compositions are mainly controlled by their mantle source, rather than by crustal contamination (Figure 4; Supplementary Table S1). The much lower Th/Nb (0.06–0.10) and higher Nb/U (42.76–71.44) ratios of the basalts studied than the average bulk continental crust (0.32 and 12.1, respectively; [67]) are inconsistent with extensive crustal contamination. Instead, the Th/Nb and Nb/U ratios of the basalts are similar to those of the average OIB (0.08 and 47.06, respectively). Minor crustal contamination is shown by the

discrimination diagrams in which these Cpx phenocrysts have alkaline to tholeiitic and calc-alkaline transitional character (Figure 10a) [92]. Moreover, the magma also displays some relative oxidation state during its crystallization and interaction with the crust (Figure 10b) [93].

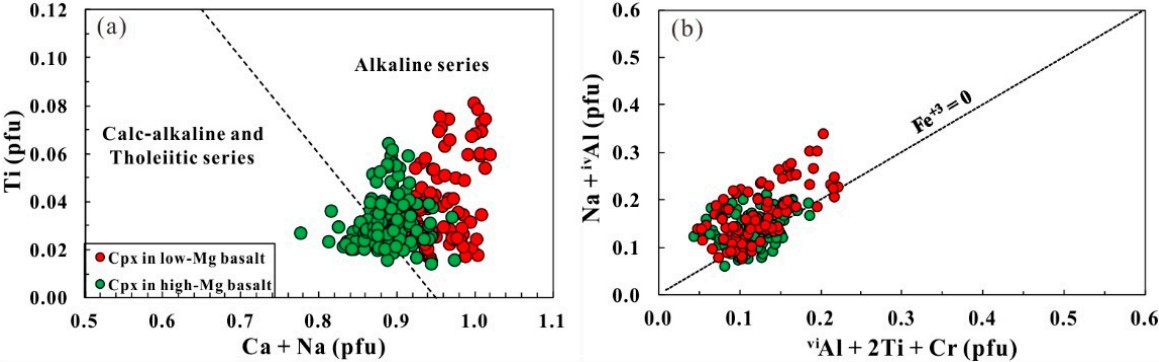

**Figure 10.** (**a**) Discrimination diagrams of (Ca+Na) vs. Ti (from Leterrier et al. (1982) [92]); (**b**) ($A^{vi}$ + Na) vs. ($Al^{vi}$ + 2Ti + Cr) (from Schweitzer et al. (1979) [93]).

Therefore, based on the study of Tao et al. (2015) and Zhang et al. (2006) [11,55], we suggest that the primary mantle-derived magma may have ascended through the deep faults formed picrites, and reached the middle crust to form the high-Mg basalts. The oscillatory and sector zoning in the Cpx phenocrysts of the high-Mg basalts indicate that the crystals experienced complex magmatic processes during the magma ascent: At the early stage, the inner part of the zoned Cpx phenocryst infers that the magma had undergone low-degree undercooling inside a deep staging chamber, and the continuous growth of Cpx sectors is compromised by multiphase magma recharge events, which may be under near equilibrium condition. These features show that the magmas may have ascended slowly along the fractures and eventually ponded in a relatively stable magma reservoir at the upper-middle crustal level. Continuous magma replenishments may have eventually led to volcanic eruptions and formed the high-Mg basalts. Some of the fractionated high-Mg basalts may have ascended up to the upper-crust staging chamber for further fractionation with minor crustal contamination, and erupted to form the low-Mg basalts or cumulated to form the mafic intrusions. The magma replenishments and ascent rate have major effect on the formation of the ELIP magma plumbing system. A schematic model of our proposed ELIP magma plumbing system is illustrated in Figure 11.

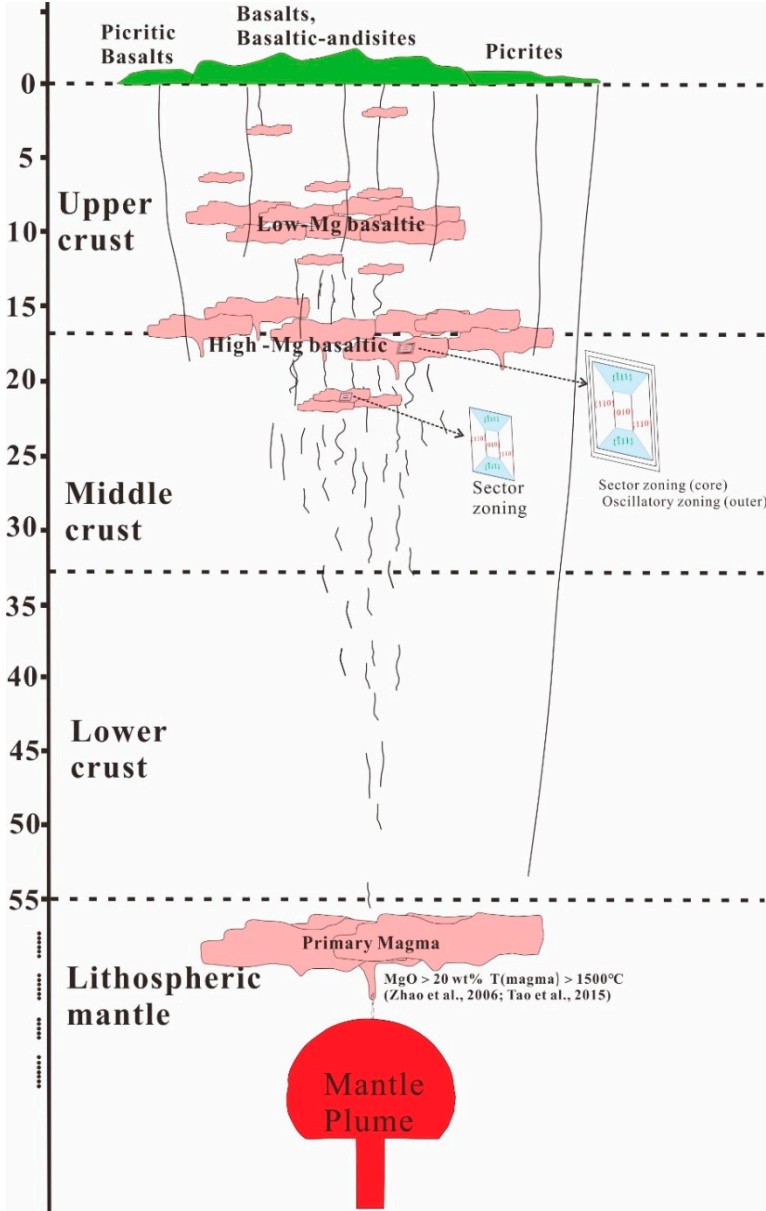

**Figure 11.** A schematic model of the magma plumbing system of the ELIP.

## 6. Conclusions

(1) Cpx-liquid thermobarometry suggests that the staging chamber(s) beneath the ELIP spread across the middle to upper crust. Cpx phenocryst crystallization for the high-Mg basalts took place mainly in the middle crust, and those for the low-Mg basalts happened in the upper crust.

(2) Similar to the giant layered mafic-ultramafic intrusions (e.g., Panzhihua intrusion in ELIP), the high-Mg basalts experienced multistage magma replenishments in the deep-staging chamber(s). This suggests that magma replenishments played a key role in generating the ELIP magma pluming system.

(3) Sector zoning of Cpx in the picrites indicate that the crystals had grown slowly under low-degree undercooling conditions. The compress growth of the crystals were also nearly-equilibrium conditions. This stable environment may induce prolonged fractionation of the high-Mg magmas to form the low-Mg basalts and/or mafic-ultramafic intrusions in the shallow staging magma chamber(s).

**Supplementary Materials:** The following are available online at http://www.mdpi.com/2075-163X/10/11/979/s1, Figure S1: Major and trace elements composition of the whole rock. (a) $TiO_2$ versus MgO; (b) $SiO_2$ versus MgO; (c) CaO versus MgO; (d) $Al_2O_3$ versus MgO; (e) Ni versus MgO; (f) Cr versus MgO; (g) Sr versus MgO; (h) V versus MgO. Figure S2: Major element composition of the Cpx phenocryst from high-Mg and low-Mg basalts. (a) $TiO_2$ versus $SiO_2$. (b) $Al_2O_3$ versus $SiO_2$. (c) $Cr_2O_3$ versus $SiO_2$. (d) $^{iv}$Al versus $Al_2O_3$. (e) Si + Mg (pfu) versus Al (pfu). (f) Mg (pfu) versus Al (pfu), Figure S3: Elemental profiles (C to D) for the zoned Cpx phenocryst of the high-Mg basalts (sample YY13-123). Profile analysis shows oscillatory zonings in the elements of Si, Mg, Cr, Ti and Al. The MgO and $Cr_2O_3$ show strong negative correction with the $Al_2O_3$, $TiO_2$ and FeO. The {010} shows $Al_2O_3$, $TiO_2$ FeO enrichment compared with the {100}. (a–h) Major elements variations along the line (from C to D), Figure S4: (a) Fe-Mg partitioning between Cpx phenocrysts and matrix. (b) Fe-Mg partitioning between Cpx phenocrysts and the matrix which extra olivine has been eliminated. (c–f) Measured and calculated values for Cpx phenocrysts components (mole fraction) from the picrite and basalt. (DiHd-diopside + hedenbergite, EnFs-enstatite + ferrosilite, CaTs-Ca-Tschermak, and Jd-jadeite. The equilibrium envelopes are marked by the standard error of estimate (SEE), which are for diopside-hedenbergite ± 0.06 SEE; enstatite-ferrosillite ± 0.05 1SEE from Mollo et al. (2013) [45]; Ca-Tshermack ± 0.08 2SEE and Jadeite ±0.04 2SEE from Putirka (1999) [84])., Table S1: The major and trace elements of the whole rocks, Table S2: The major elements of the matrix, Table S3: Major elements of the Cpx and Ol phenocrysts, Table S4: Results of the P-T calculations.

**Author Contributions:** Conceptualization, J.-H.H. and J.-W.L.; formal analysis, J.-H.H.; investigation, J.-H.H., J.-W.L., and B.-S.S.; resources, J.-H.H. and J.-W.L.; writing—original draft preparation, J.-H.H., J.-W.L.; writing—review and editing, J.-H.H. and T.S.; visualization, J.-H.H.; supervision, J.-W.L.; funding acquisition, J.-H.H. All authors have read and agreed to the published version of the manuscript.

**Funding:** This study was funded by the Talent start-up fund of Guiyang University (2019039510821)-Research on the magmatic process and its genesis mechanism of Tengchong volcanic rocks, and the Priority Research Program (B) of Chinese Academy of Sciences, Grant No. XDB18000000.

**Acknowledgments:** We gratefully acknowledge the senior engineer Wen-Qin Zheng and Xiang Li of the Institute of Geochemistry, Chinese Academy of Sciences for their guidance in EPMA analysis work. We thank Kai-Yuan Wang for their handling of the manuscript and editorial input. We would like to thank Sheng-Hua Zhou and Jian Kang for their assistance during fieldwork.

**Conflicts of Interest:** The authors declare no conflict of interest.

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
