# Peer review of "Magma Plumbing System of Emeishan Large Igneous Province at the End-Permian: Insights from Clinopyroxene Compositional Zoning and Thermobarometry"

_minerals, doi:10.3390/min10110979_

Round 1

Reviewer 1 Report

Dear Editor,

Below is my review for the manuscript “ The magma plumbing system of Emeishan large 2 igneous province insight from the compositional 3 zoning and thermobarometric calculation of the 4 clinopyroxene phenocrysts” by Hu et al.

This manuscript deals with the zoning pattern of the cpx mineral from basalts and picrites and based on the zoning pattern in combination with the mineral and phenocryst composition the authors investigate to understand the magma storage and plumbing system. This manuscript provides a better understanding of the magma generation and storage system in particularly to the Emeishan LIP. Therefore, this manuscript fits within scope of this journal. The conclusions are reasonable and well supported. So my suggestion is to accept with major/minor correction. I leave this up to the editor.

Below are my comments:

 I suggest the manuscript will need to be read by an English speaking person. There are many typos and construction they need to be fixed before publication and can be easily fixed.

Analytical techniques are Ok, with the exception of technique they have used to determine the groundmass composition. Please my line by line comments for details.

Line 134-139:  This is not clear. They mention two different settings for the phenocryst major element analysis with two different spot size. One is for Cpx and it is not clear other one for which minerals?

Line 121-127: If I understand correctly you have gone through 2-3 kg of rock powder to pick out the phenocryst? Somehow this does not seem realistic. In addition, it is inevitable that you have missed many phenocrysts. Therefore, the phenocrysts will have a huge impact on the matrix composition!! What would be the uncertainty related to that.  

Line 285: “error of estimate is 45K” -what is 45K? this is not clear.

Line 301-305: What are the dotted lines in the figure 11? Are they some sort of equilibrium composition? Why there are four lines? Did you do the modal abundance calculations? How did you do that? Similar to olivine there could be some cpx in the groundmass!! In figure 11b, there is still a lot of scatter- there are data points that plot above and below the equilibrium condition, if these lines indicate the equilibration. This observation is  very serious of the basaltic composition in comparison to picrites.

Line 304: “ These modal amounts adding the those of olivine phenocrysts picked out under the binocular microscope are less than observed modal amounts of olivine in the whole rocks”- Not sure what does that mean- why less?  you would expect they will be equal!

Reviewer 2 Report

See attached file

Reviewer 3 Report

I read the manuscript entitled “The magma plumbing system of Emeishan large igneous province insight from the compositional zoning and thermobarometric calculation of the clinopyroxene phenocrysts” with interest but the paper is not always well written. There are some problems with the English and I suggest the paper is read by an English mother tongue before resubmission. The main issues I find in this manuscript are in the discussion and conclusion which I believe are a bit poor. Reference to previous literature should be added as well as a comparison with similar systems.

Figures and tables are ok but there are a lot of diagrams, perhaps they need to be reviewed to verify that they are all necessary. To my opinion there are some revisions the paper needs to go through before publication and there are some issues that I would like to suggest to fix before resubmission.

Here are a couple of main points together with the ones to be found noted on the paper.

  • References need to be checked
  • The manuscript discussion section need to be reviewed and made reader friendly. It is very intricate and complicated to follow.

Specific comments and correction can be found on the attached PDF File.

Reviewer 4 Report

Review for the manuscript entitled:

“The magma plumbing system of Emeishan large igneous province insight from the compositional zoning and thermobarometric calculation of the clinopyroxene phenocrysts”

by

Jun-Hao Hu, Jing-Weng Liu, Bai-Shun Shi, and Tao Song

 This manuscript presents several bulk rock and matrix analyses and focusing clinopyroxene composition in flood basalts of the Late Permian Emeishan large igneous province. This research deals with the reconstruction and magma dynamics of a multistage magma reservoir. Since I'm being a big fan of classical EPMA geothermobarometry and I find the data generally of good quality, I would like to see this material ultimately published.

However, after careful consideration, if the manuscript in its present form could be modified to eradicate its unfortunately pervasive technical and structural problems, I concluded that even major revisions would fall short in bringing this manuscript to the standard expected for a journal targeting an international audience. My conclusion is therefore to recommend rejection with the options of re-submission. Before resubmitting the manuscript to the MDPI Minerals journal, the authors should improve the structure of the manuscript (parts from discussion needs to be shifted into introduction or result, see annotated manuscript) and should rewrite some chapters (please watch out the mode of expression). It would be a great benefit to strengthen the title of targeting an international audience. I am not from China and I would like to have information about the geological era - It is quite interesting to know that the ejection of large-volume lava flows in the Emeishan area could be linked to the Late Permian mass extinction. Please find general comments below, and more detailed in an annotated word file.

Unfortunately, the following major issues contributing to the problems with this paper:

  • Introduction

First, you have to tell the reader about the importance of your study by referring to other LIPs for example (this will shift your manuscript immediately out of the local context)! I give some examples in the annotated manuscript.

Since intrusions of Baima, Hongge, Panzhihua, and Taihe forms important Fe-Ti-V-oxide ore deposits and those of Baimazhai, Limahe and Jinbaoshan are economically valuable Ni - Cu - PGE sulfide deposits in the Emeishan large igneous province (see your last paragraph in Chapter 2 Geological background), the manuscript contributes to the understanding of magmatic ore deposits! Fluids of the magma system influence ore-precipitation - this is missing in the purpose. (see Ernst & Jowitt, 2013 and references therein).

See:

Ernst, R. E., Jowitt, S. M. (2013). Large igneous provinces (LIPs) and metallogeny. Society of economic geologists special publication, 17, 17-51

4) Results

4.1 Whole - rock and matrix geochemical compositions

Please check your dataset again.

Following the IUGS classification, a picrite is defined by SiO2 < 45.3 wt. % (see TAS classification, Le Maître, 1984; Le Bas et al., 1986). Strangely, your picrites have SiO2-contents between 45.65 and 47.50 wt%.

Solely, the matrix analysis of sample YY13-123 (Shinan, electron supplement table 2) fulfills the condition SiO2 < 45 wt.%. However, the concentration of alkalis (Na2O+K2O = 0.82 wt.%) and high Mg-content (20.67 wt.%) refer to a komatiitic composition (see Le Bas, 2000; Kerr and Arndt, 2001). This means that this important chapter needs to be rewritten.

Kerr, A.C., Arndt, N.T., 2001. A note on the IUGS Reclassification of high-Mg and picritic volcanic rocks. Journal of Petrology 42 (11), 2169-2171.

Le Bas, M.J., Le Maître, R.W., Streckeisen, A., Zanettin, B., IUGS subcomission on the systematics of igneous rocks, 1986. A chemical classification of volcanic rocks based on the total alkali-silica diagrid. Journal of Petrology 27 (3), 745-750.

Le Bas, M.J., 2000. IUGS reclassification of the high-Mg and picritic volcanic rocks. Journal of Petrology 41, 1467-1470.

Le Maître, R.W., 1984. A proposal by the IUGS Subcommission on the systematics of igneous rocks for a chemical classification of volcanic rocks based on the total alkali silica (TAS) diagrid. Australian Journal of Earth Sciences 31, 243-255.

You should not write all the compositional data here! Refer to Mg-number (Mg# = 100 x Mg/Mg+Fe2+), then refer to Ti-concentration.

The work of Gibson et al. (1995) defined high- and low-Ti basalts, with TiO2 > 2.5 wt. % and TiO2 < 2.5 wt. %, respectively. You also mentioned this in the chapter of the geological setting (Line 76.)

In this case, you would have both types!

Gibson, S.A., Thompson, R.N., Dicking, A.P., Leonardos, O.H., 1995. High-Ti and low-Ti mafic potassic magmas: Key to mantle plume lithosphere interactions and continental flood-basalt genesis. Earth and Planetary Science Letters 136, 149-165.

Mention that the two definitions revealed disparities!

Following your trace element analyses, the Ti/Y ratio shows throughout > 500. It is always good to have a critical eye on your results!

4.2 Phenocrysts compositions

First, please classify your clinopyroxene phenocrysts. Your analyzed pyroxene belongs to the augite-diopside-solid solution series. Just plot the pyroxene classification quadrilateral. Remove all unnecessary information (and diagrams) – create valuable diagrams! (see annotated manuscript!). My suggestion is to refer to clinopyroxene zonations and highlight only significant variations in crystal chemistry. Parts of paragraph 5.1 Petrogenetic implications from zoned Cpx phenocrysts needs to be shifted into the results part. (see annotated manuscript!). This important chapter needs to be rewritten.

5) Discussion

5.1 Thermobarometry

Please include also the clinopyroxene-only thermobarometry of Nimis & Taylor (2000) (Eq. 32d in Putirka (2008)). It is an H2O-independent thermobarometer, which has of course advantages and disadvantages. However, it contributes to your discussion.

Nimis, P., Taylor, W.R., 2000. Single clinopyroxene thermobarometry for garnet peridotites. Part 1 Calibration and testing on Cr-in cpx barometer and an enstatite in cpx-thermometer. Contribution Mineralogy and Petrology 139, 541-554.

Putirka, K. D. (2008). Thermometers and barometers for volcanic systems. Reviews in mineralogy and geochemistry, 69(1), 61-120.

Please shorten the part about the exchange coefficient (KD).

After geothermobarometry, I also recommend plotting the discrimination diagrams for calcic clinopyroxene! Your clinopyroxene reveals an alkaline to tholeiitic and calc-alkaline character, which indicates crustal contamination! This combined with your trace element variation contributes to the final sketch (Fig. 14) manuscript:

Please see:

Leterrier, J., Maury, R.C., Thorron, P., Girard, D., Marchal, M., 1982. Clinopyroxene composition as a method of identification of the magmatic affinities of paleo-volcanic series. Earth and Planetary Science Letters 59 (1), 139-154.

Schweitzer, E., Papike, J., Bence, A., 1979. Statistical analysis of clinopyroxenes from deep-sea basalts. American Mineralogist 64, 501-513.

Artwork and diagrams

Figures in the text should always support your interpretations! Since I saw pointless element maps and diagrams in your manuscript, I must kindly ask for checking the necessity of figures. I welcome new figures before resubmission (see the annotated manuscript and comment below).

Figure 1 and 2:

The simplified geological maps of the studied area are in general of good quality. However, the authors can merge these figures 1 and 2:

- the number of figures can be reduced, so you have space for more valuable figures

- information is clearer and more suited to the reader

Figure 3:

Chromite is a bit difficult to see within the figure. Is SEM-BSE-image possible?

Figure 4:

Please check the necessity of diagrams. Why not using a TAS diagram? This should make clear that your analyses do not show any picritic composition and make the interpretation questionable! - Solely, one matrix analysis revealed a komatiite composition; sample YY13-123 [Shinan, electron supplement table 2]).

You should present diagrams for trace element variation normalized to primitive mantle and chondrite (see Sun & McDonough, 1989), since you analyzed them. It can be helpful for interpretation if you add the compositions of continental crust (see Taylor & McLennan, 1995). Together with discrimination of clinopyroxene, the crust-melt interaction can be tracked in this flood basalt province.

Sun, S. S., & McDonough, W. F. (1989). Chemical and isotopic systematics of oceanic basalts: implications for mantle composition and processes. Geological Society, London, Special Publications, 42(1), 313-345.

Taylor, S. R., & McLennan, S. M. (1995). The geochemical evolution of the continental crust. Reviews of geophysics, 33(2), 241-265.

Figure 5:

Please check the necessity of diagrams. It is important to classify the clinopyroxene phenocrysts (diopsid-augite solid solution series), which I haven’t read within the manuscript. Why not just plotting a pyroxene quadrilateral?

Figures 6 and 7:

Figures 8, 9, and 10:

These figures belong to the results part. They are of very good quality.

Figures 11 to 12:

Please shift these figures to the electronic supplement.

Figures 13 to 14:

These figures are excellent. However, you can add the clinopyroxene-only thermobarometry of Nimis & Taylor (2000) (Eq. 32d in Putirka (2008)).

Nimis, P., Taylor, W.R., 2000. Single clinopyroxene thermobarometry for garnet peridotites. Part 1 Calibration and testing on Cr-in cpx barometer and an enstatite in cpx-thermometer. Contribution Mineralogy and Petrology 139, 541-554.

Putirka, K. D. (2008). Thermometers and barometers for volcanic systems. Reviews in mineralogy and geochemistry, 69(1), 61-120.

Additional Figures:

I also recommend plotting the discrimination diagrams of Schweitzer et al. (1979) [AlVI + Na vs. AlVI + 2Ti + Cr] and Leterrier et al. (1982) [Ca+Na vs Ti; Ti vs. Altot) – the authors might be surprised that they can tell a story of magma oxidation state and crustal contamination during crystal solidification. It would support the plumbing magma system model (Fig.14). This is helpful for an impactful discussion and the manuscript will get considerable attention:

Your clinopyroxene reveals an alkaline to tholeiitic and calc-alkaline character, which indicates crustal contamination.

Leterrier, J., Maury, R.C., Thorron, P., Girard, D., Marchal, M., 1982. Clinopyroxene composition as a method of identification of the magmatic affinities of paleo-volcanic series. Earth and Planetary Science Letters 59 (1), 139-154.

Schweitzer, E., Papike, J., Bence, A., 1979. Statistical analysis of clinopyroxenes from deep-sea basalts. American Mineralogist 64, 501-513.

If the authors can attend to these concerns, this paper will be a much more impactful contribution to the understanding of magma reservoirs beneath continental flood basalt provinces. And I strongly recommend a re-submission of an improved manuscript.

regards

Round 2

Reviewer 4 Report

Dear authors,

please find attached my comments to your submitted manuscript.

Regards

Author Response

Dear reviewer

Thank you.
